# Exploring the Associations between Autistic Traits, Sleep Quality and Well-Being in University Students: A Narrative Review

**DOI:** 10.3390/healthcare12202027

**Published:** 2024-10-11

**Authors:** Devangi Lunia, Andrew P. Smith

**Affiliations:** Centre for Occupational and Health Psychology, School of Psychology, Cardiff University, 63 Park Place, Cardiff CF10 3AS, UK; smithap@cardiff.ac.uk

**Keywords:** autistic traits, broader autism phenotype, sleep quality, well-being, university students, ADHD, depression, anxiety

## Abstract

While research on autism spectrum disorder (ASD) has been growing, not enough research has been conducted to understand the impact of autistic traits and the broader autism phenotype (BAP), especially on the sleep quality and well-being of university students. The focus of this paper is to review the existing research on this topic and identify the key areas of interest for future research, presented in the form of a narrative review. While the review identifies the need for research on the topic, it also identifies other factors such as gender, age, culture, and internet and smartphone use that may have an impact on the relationship between autistic traits, sleep quality and well-being in university students. The review also identified the importance of using a larger sample size, appropriate measures, especially for quantifying autistic traits, and an appropriate analysis strategy involving a multivariate analysis.

## 1. Introduction

Our understanding of autism spectrum disorder (ASD) has changed due to its complex nature. ASD is a neurodevelopmental disorder that causes impairments in social communication and repetitive behaviours that vary in severity, ranging from low support needs to high support needs. Its severity and scope depend on varying intellectual and language abilities, significantly impacting the individual’s social communication capabilities [1]. However, people who do not meet the threshold for diagnosis of the disorder, i.e., those that fall under the broader autism phenotype (BAP), may display autistic traits, which comprise impairments in social interactions, communication, social camouflage, cognitive rigidity, repetitive behaviours, and sensory sensitivity [2,3]. These traits were identified based on the diagnostic criteria of ASD, which involves atypical responses in childhood to sensory stimuli, emotional, motor skills, daily activities and play and difficulties in learning language and speech, and in imitating others, with sensory sensitivity being the most recent addition as seen in the Diagnostic and Statistical Manual of Mental Disorders fifth edition, revised (DSM-V-TR) [4,5]. These autistic traits have been known to impact mental well-being significantly.

Another major problem is the conceptualisation of well-being. Well-being is a free-standing and multifaceted social construct that is difficult to define and quantify [6]. However, the Well-being Process model [7] encompasses its components effectively, which will help structure this project. This model was based on the Demand-Resources-Individual-Effects (DRIVE) (as displayed in Table 1) [8,9], which involves the measurement of general characteristics (positive: social support; negative: demands), appraisals (positive: life satisfaction; negative: perceived stress), individual characteristics (negative: negative coping; positive: positive personality) and outcomes (positive: happiness; negative: anxiety and depression). This model requires measuring various factors that impact well-being, which provides a holistic approach to quantifying them.

While the DRIVE model helped in quantifying well-being by identifying the factors impacting it in a workplace, the Well-being Process model can be applied to both workers and students. This model helps in identifying the well-being predictors and outcomes so a correlation between the two can be established. Based on the table above, social support, demands, positive personality and negative coping are predictors, and happiness, anxiety and depression are outcomes. Life satisfaction and perceived stress act as mediators in the relationship between the predictors and outcomes [7]. These components are vague, but they are measured in depth through the Well-being Process Questionnaire, which was developed based on this model.

When considering well-being, it is also essential to consider the role of co-morbidities of ASD, two of which have been mentioned in the well-being process model: depression and anxiety. Depression and anxiety are common co-morbidities for autistic people, with pooled prevalence estimates of 11% and 20%, respectively, as found through a systematic review of 96 studies conducted between 1993 and 2019 [10]. However, other co-morbidities, such as attention deficit hyperactivity disorder (ADHD), with a pooled prevalence estimate of 28% [10], should also be studied, and their impact on mental well-being should be understood. While ASD and ADHD are distinct disorders, there is some overlap between the two, explaining their co-morbidity.

Sleep quality is another factor that impacts well-being. It was defined as personal satisfaction with the overall sleep experience, which includes sleep efficiency, latency, sleep duration and wake-after-sleep onset (WASO) [11]. Measures include how quickly an individual can fall asleep, if they can stay asleep, the number of hours of sleep they have, and how rested they feel after sleep. There are many antecedents (such as age, body mass index, stress, depression and anxiety) and consequences (such as strained social relationships, irritability, fatigue and exhaustion) of poor sleep quality, which may impact well-being significantly [11]. Some of these factors, especially strained social relationships, appear part of the BAP.

Other factors, such as irritability and fatigue, may lead to impairments in adults and youths with ASD. One study in 2023 found significant genetic associations between ASD and fatigue from summary statistics from the U.K. biobank consisting of 108,976 responses, suggesting that autistic people are more prone to feeling fatigued [12]. Previously, a study in 2021 found that children with ASD often display irritability through high levels of angry moods and temper tantrums [13]. A causal mechanism for this is associated with problems in communication or interruption of repetitive behaviours and sensory activities. While considering fatigue, university students often reported fatigue due to academic stress and poor sleep quality due to negative coping styles, workload stress and disturbed surroundings [14]. However, this has not been considered in relation to autism. Due to the overlap between ASD and BAP traits, it is essential to consider the impact of fatigue and irritability on well-being and its correlation with poor sleep quality in adults displaying autistic traits.

## 2. Methodology

### 2.1. Design

A narrative review was conducted to explore the gaps in research in this area and how they can be filled. All articles used in this review were published in English. One must now consider the limited literature relating to the above topics. There is a much smaller amount of literature on autistic traits than autism, with autism producing around 78,000 citations and autistic traits producing around 3000 citations on PubMed. 1416 citations are produced when “autism” and “well-being” are searched, 2331 citations for “autism” and “sleep”, and only 61 citations for “autistic traits” and “sleep”. Combining the three search terms gives a similar picture, with “autism”, “sleep”, and “well-being” producing 75 citations, and “autistic traits”, “sleep” and “well-being” producing only 4 citations. However, an even more significant gap in the literature is observed when studies of university students are examined, with “university students” and “autistic traits” showing 182 citations, “university students”, “autistic traits” and “well-being” showing 12 citations, “university students”, “autistic traits” and “sleep” showing only 3 citations, and “university students”, “autistic traits”, “well-being” and “sleep” only shows 1 citation—and that is not a relevant study. Research exploring the impact of sleep quality on the mental well-being of autistic people has been growing, but more focus is required on this. Hence, this review aims to provide the background for empirical research.

### 2.2. Search Strategy

Google Scholar was used, and only research conducted between 2019 and 2024 was considered to keep the findings from these papers relevant to the current study. ‘Autism’, ‘ASD’, ‘Autistic traits’, ‘Sleep quality’ and ‘Mental Well-being’ were the search terms used. However, most studies appeared to study children or adults, so ‘University students’ was also used. Ultimately, ‘Autism’ and ‘ASD’ were removed from the list of search terms to explore the difference between autistic adults and adults under the BAP in terms of well-being. This was explicitly in the symptoms experienced by each group and the support they received for these symptoms, which determined how these individuals managed their symptoms daily. Finally, it was also essential to consider co-morbidities for ASD as this contributes to mental well-being vastly. Hence, ‘Co-morbidity’, ‘Depression’, ‘Anxiety’ and ‘ADHD’ were added to the list of search terms as these are most relevant to the current study.

### 2.3. Analysis Method

Based on Sukhera’s (2022) key steps for conducting a narrative review, studies are explained and interpreted at the same time [15]. Hence, each study that adhered to the inclusion and exclusion criteria was described, and its findings were explained. All interpretations of findings have been supported by other research conducted in the field. Then, the strengths and limitations of each study were explored, and their importance in explaining a rationale for conducting research on this topic was discussed, along with what aspects of the studies will be adapted and how the limitations are accounted for in future empirical studies.

## 3. Results and Discussion

### 3.1. Impact of Autistic Traits on Well-Being

When considering the relationship between quality of life and autistic traits, much research focuses on people with an ASD diagnosis, with not enough research on people who may fall under the BAP. However, studies are starting to highlight the associations between autistic traits and quality of life, suggesting that research in this area is required to effectively recognise and support individuals who may be struggling. Lawson and colleagues (2020) investigated the predictors of quality of life in 244 autistic and 165 non-autistic individuals aged 15 to 80 years old [16]. They reported that in both autistic and non-autistic samples, demographics (age, gender, living situation, education status and employment status), autistic traits, vocation, mental health, and physical health significantly accounted for variance in Physical Quality of Life; autistic traits, vocation, mental health and well-being accounted for variance in Social Quality of Life and demographics; autistic traits and mental health significantly accounted for variance in Environmental Quality of Life.

Autistic traits, mental health and well-being significantly accounted for variance in Psychological Quality of Life in the autistic sample, with the addition of demographics and vocation in the non-autistic sample. When considering gender and age, the researchers reported that autistic males and older autistic adults had poorer social quality of life, which contrasts with the findings of other researchers that they reviewed. Overall, it appeared that autistic traits significantly impacted all four domains of quality of life considered in this study. Similarly, Oakley and colleagues (2020) measured the quality of life in 573 individuals between 6 and 30 years old [17]. They reported that 344 individuals displayed autistic traits, suggesting that there may be more people with autistic traits than previously thought. It was found that individuals with autistic traits scored significantly lower on the quality-of-life scale as compared to those who did not display autistic traits.

This emphasises the importance of research in this population as individuals who display the BAP often have their symptoms unrecognised, which can negatively impact their quality of life as their healthcare needs may be unmet. Furthermore, these traits appeared to be more prominent, and to have significantly impacted children and adolescents more than adults due to decreased overall satisfaction and school achievement. The authors highlight how the complexity of school settings, in terms of social and sensory environments, can be challenging for young people on the autism spectrum. These implications should also be considered in university students as they are exposed to similar settings and dissatisfactions with increased stress due to workload and the need to adapt to new environments such as a new city living in shared accommodations instead of living with parents.

### 3.2. Research on ASD University Students

Autistic traits and well-being at university have also been studied. Flegenheimer and Scherf (2021) conducted a systematic review of 62 ASD college student studies to explore how college impacts development in ‘emerging adulthood’ (involving individuals between 18 and 25 years old) in autistic adults compared to their neurotypical peers [18]. ASD students self-reported having an intense interest in their subject. However, it was also found that they have more difficulty focusing on the college environment, suggesting ASD students have to work harder to focus and achieve similar performance levels as their neurotypical peers. ASD students also found it more challenging to acquire functional adaptive behaviours, leading to higher academic stress levels, daily life (personal care, money, time management) and socialisation than their neurotypical peers. Time management was challenging for ASD students because of their impaired flexibility due to a ‘symptomatic preference for routines’. Reports on sleep hygiene were inconclusive as some studies found that ASD students were less likely to report being sleep deprived, while another study found that ASD students reported quality of sleep to be a moderate or significant concern.

ASD students (and ADHD students in some studies) also found it challenging to attain social interactions and friendships even though they reported a desire for them, leading to higher levels of loneliness and low confidence in their social skills. ASD students also found college to be an ‘overwhelming sensory experience’ causing a decline in academic performance. Lane, Leao and Spielmann (2022) found in their scoping review that autistic children and adults reported high sensory sensitivity, and sensory sensitivity may not be influenced by age, autism severity, medications and intellectual abilities [19]. They also reported that visual sensory sensitivity impacted sleep quality in adults, and in children, it was poor auditory filtering and taste/smell differences. However, the relationship between sensory sensitivity and sleep is complex and multifaceted, as results were inconsistent when conducting the review.

Flegenheimer and Scherf (2021) also reported that ASD students felt a greater need for ‘masking’ their symptoms while at college, which made social interactions’ exhausting, challenging and less rewarding’ [18]. This aligns with the findings of other studies on camouflaging and masking behaviours, such as the study conducted by Alaghband-Rad, Hajikarim-Hamedani and Motamed (2023). They conducted a systematic review of 16 articles and concluded that autistic adults tend to camouflage in social situations to fit into neurotypical communities, with autistic females more likely to engage in ‘camouflaging’ than autistic males [20]. While not many studies explored the reasons for ‘camouflaging’, they reported its negative impacts such as anxiety, depression and exhaustion, which negatively impact their daily life aspects such as employment, university graduation, relationships, financial status and overall quality of life. Overall, it can be concluded that college/university life can be extremely overwhelming and challenging for autistic individuals compared to their neurotypical peers.

Associations between autistic traits and mental health have also been investigated. Campbell and colleagues (2022) conducted a systematic review of 31 studies published between 2010 and 2020 to establish the factors that impact mental health in 17,476 university and college students in the U.K. [21]. They found that LGBTQ students and students with adverse childhood experiences and personal or family history of mental illnesses were at significantly greater risk of depression, anxiety, suicidal behaviour, self-harm and low levels of well-being. ‘Autism’ as a factor had a significant correlation with poor social problem-solving skills and depression. However, findings on ‘age’, ‘gender’, and ‘ethnicity’ as factors were inconsistent among the reviewed studies. ‘Stress’ was also explored as a trigger factor as university students experience high levels of stress due to academic workload and new social and academic environment. Other university-related factors identified in the review were financial anxieties (which caused long working hours) and accommodation factors, negatively affecting students’ mental health.

In the above review, problem drinking/alcohol dependence, harmful use of illicit drugs, poor sleep quality and increased consumption of unhealthy foods were identified as the outcomes of factors identified. This study highlights a correlation between sleep quality, well-being and autism; however, more research is needed to focus on the relationship between these factors and how they may impact the lives of university students, as the factors that the researchers identified are common among university students which negatively affect their mental health. It may also be necessary to consider age and gender as factors to add to existing findings and draw conclusions on their impact on the relationship between the three factors being studied. This aligns with Lawson and colleagues’ (2020) findings [16], which showed that sleep quality and autistic traits were significant predictors of physical quality of life in autistic individuals.

In addition, Zhang and colleagues (2022) studied 1928 university students from five universities in China to find out how lifestyle, sports habits and mental health impacted sleep quality [22]. They evaluated 22 potential risk factors, some of which included ‘having a habit of midday rest’ and ‘mental health status (depression, anxiety and stress)’, while other factors considered demographics and preference in food. Only considering the factors relevant to the present study, the researchers found that older age, year of study, midday rest, anxiety and stress were significantly correlated with high rates of poor sleep quality. However, it is unknown if the same result would be obtained in people with autistic traits, as ASD and BAP were not considered in this study but are the focus of the present research.

To add to the findings of this study, Sivertsen and colleagues (2021) studied the prevalence of delayed sleep–wake phase disorder (DSWPD) in 50,054 Norwegian university students. They also considered self-reported mental disorders such as anxiety disorder, depressive disorder, ADHD, and suicidality students and compared the students with and without DSWPD [23]. The estimated prevalence of DSWPD was 3.3%, which was like previous studies conducted on adolescents. While the previous study found that females had a higher prevalence of DSWPD than males in late adolescence, this study found that males had a higher prevalence than females when they reached young adulthood. The researchers also found that students with DSWPD were significantly more likely to show mental health symptoms, somatic symptoms, loneliness and alcohol problems. More specifically, students with DSWPD were more likely to have a diagnosis of anxiety disorder, depressive disorder and ADHD, and reported more suicide attempts and suicidal thoughts than students without DSWPD. While the present study does not focus on DSWPD, the findings of this study indicate the impact sleep quality may have on the well-being of university students and show that university students are more likely to have poor sleep quality or sleep problems than adolescents. It is also important to understand how these findings may differ with the presence of autistic traits.

Several research papers highlight the experience of college and university students on the autism spectrum or the BAP and the struggles they face as compared to their neurotypical peers, which highlights the importance of research in this area. McKenney (2022) compared the experiences of 60 non-autistic and 36 autistic students from four universities (two from New Jersey, USA and New York, USA each) to establish some predictors of emotional health in the student’s first semester in college [24]. They found that negative repetitive thinking and depressive attributional styles were significantly predictive of ‘sadness scores’, and negative repetitive thinking was also significantly correlated with anxiety symptoms. Autistic students were more likely to experience negative repetitive thinking, social dissatisfaction, heightened depressive attributional style, and depressive symptoms than their non-autistic peers. This suggests that autistic students experience more depression and anxiety throughout their university life compared to their neurotypical peers, specifically due to autistic traits such as repetitive behaviours and problems with social interactions.

Similarly, Anderson, Carter and Stephenson (2019) conducted an online survey of 102 university students with ASD across Australia and New Zealand to determine the factors that impact university and support satisfaction [25]. They considered gender, age, mental health, strength and difficulties and academic performance. They found that many students took time off before starting university, which was similar to the findings in a previous study they discussed, which involved Finnish and Australian students. 18% of the participants also reported switching from full-time to part-time, which, the researchers suggested, might be because ASD students struggle more with being at university than the general student population. They also reported that sensory sensitivity was a big concern for most students, which affected their ability to study or cope in university, which aligns with the findings discussed above [17,18,19]. In terms of mental health, anxiety and depression were a moderate or great concern for most participants, and females showed higher rates of self-harm and suicidal ideation. Loneliness and poor socialisation were found to be correlated with high rates of depression, anxiety, low self-esteem and life satisfaction.

### 3.3. Factors That Affect Well-Being in People with Autistic Traits

One factor that may explain the challenges faced by people with autistic traits is Alexithymia, which, simply put, is defined as a difficulty in identifying and describing feelings and can impact social-emotional processing [26]. Vaiouli and Panayiotou (2021) studied the correlations between Alexithymia and autistic traits in 275 college students in Cyprus [27]. There was a significant positive correlation between autistic and alexithymic traits and autistic traits and emotion regulation difficulties, but a negative correlation between autistic traits and social skills. Overall, difficulty in describing feelings can make emotional sharing, communication, and support-seeking difficult. Identifying feelings can negatively impact emotional awareness and acceptance, and externally oriented thinking can lead to avoidance of internal experiences, which may cause poor emotional regulation. These findings can explain the amount of social dissatisfaction faced by people with ASD or autistic traits, which may lead to higher anxiety and depression symptoms.

Another explanation for this could be the relationship between autism symptoms, executive functioning and academic progress, which was studied by Dijkhuis and colleagues (2020), involving 54 young adults with an average age of 22.5 years. More specifically, they studied social awareness, social communication, social motivation and autistic mannerisms using the Social Responsiveness Scale for Adults (SRS-A), alongside studying intelligence, academic progress and executive functioning, specifically daily executive functioning such as self-regulation problems and performance execute functions such as mental flexibility and working memory [28]. There were significant correlations between academic progress and SRS-A scores, academic performance, and daily executive functioning, specifically planning/organising, initiating, task monitoring, and working memory. Overall, higher autism scores and poorer daily executive functioning were associated with poorer academic performance. This provides a reason for the struggles that people with ASD and in the BAP face, which is the impairment in executive functioning due to autism and autistic traits that make it difficult for these individuals to excel and fit in in university settings.

Similar results to McKenney’s (2022) study were reported in a study by Kurtz and colleagues (2023), who studied the role of the BAP in anxiety and depression in 2976 college-aged adults and focussed especially on non-Hispanic White (2791) and Black (185) participants [24,29]. The individuals completed the Broad Autism Phenotype Questionnaire (BAPQ) and Patient Health Questionnaire (PHQ-9), and the researchers found that BAPQ scores and anxiety and depression symptoms were significantly correlated. While both race groups showed a significant correlation, the correlations were higher in Black samples than in non-Hispanic White samples, suggesting that Black individuals with autistic traits experienced more anxiety and depression symptoms. Even if race was excluded as a factor, the researchers found that participants with a high BAPQ score also experienced more anxiety and depression symptoms. This suggests that traits such as ‘social behaviour’, ‘stereotyped-repetitive behaviour’ and ‘communication’ significantly impacted the individual’s mental health. While it was reported that the Black participants’ mental health and psychological well-being were negatively affected by race-related stressors, the researchers have highlighted a possible correlation between race-related stressors and autistic traits that contribute to their experience of anxiety and depression symptoms.

Another study by McLeod, Meanwell and Hawbaker (2019) studied 3073 students across 14 institutions across Indiana, USA, where 95 students were found to be ‘on the spectrum’, 804 students with other disabilities and 2174 neurotypical students. Students ‘on the spectrum’ reported lower levels of belonging and quality of social relationships, higher rates of physical and verbal bullying, poorer physical and mental health, and higher rates of depression and anxiety than neurotypical students [30]. Interestingly, the researchers also reported that students ‘on the spectrum’ were more likely to be international students than neurotypical students, suggesting cultural differences in how certain behaviours may be perceived. While the present study does not consider cultural differences, it is important to note if the well-being of students with ASD or in the BAP is any different between China, the USA, Australia, New Zealand, and the U.K. In addition, to elaborate on the impact of stress caused by the academic workload on the well-being of university students, the researchers found that neurotypical students reported a higher GPA, fewer academic challenges, and engaged in more collaborative learning, internships, field placements and other career-oriented academic experiences than students ‘on the spectrum’. This suggests that academic workload negatively impacted students with ASD and potentially those with autistic traits.

Some studies on university students who display autistic traits have been conducted. Dell’Osso and colleagues (2022) studied 2141 students at the University of Pisa to establish the prevalence of autistic traits and camouflaging behaviours in the sample. They measured seven domains within the adult autism subthreshold (AdAS) spectrum, namely ‘childhood/adolescence’, ‘verbal communication’, ‘nonverbal communication’, ‘empathy’, ‘inflexibility and adherence to routine’, ‘restricted interested and rumination’ and ‘hyper/hypo reactivity to sensory input’ [31]. They found that university students display a high frequency of autistic traits, especially in universities with entrance exams and ‘ultracompetitive study paths’ such as Applied Sciences, Pure Sciences and Humanities. It was also reported that students in the Pure Sciences field had the highest score on the camouflaging autistic traits questionnaires (CAT-Q), and Humanities was second on the list. While they consider a wide range of fields in the university to compare the frequencies of autistic traits between them, it is important to consider how this frequency can vary within a field, in this case, the Humanities (more specifically Psychology) and consider factors that may cause individuals to camouflage more than others, for example, the year of study which was found to be of importance [22].

Additionally, it was found that males scored higher than females on all the AdAS Spectrum domains except ‘hyper/hypo reactivity to sensory input’. The researchers did not find an influence of gender on the CAT-Q scores, suggesting that males and females have equal levels of camouflaging. This is interesting as other research has found that autistic females are more likely to camouflage than autistic males. Milner and colleagues (2023) studied the gender differences in camouflaging, considering diagnosed autism, high autistic traits and low autistic traits in 105 participants aged between 20 and 25 years [32]. They also used the CAT-Q to measure the differences and found that females with an autism diagnosis had higher scores than males. However, when undiagnosed on both levels of autistic traits, both sexes reported the same level of camouflaging. This highlights the difference between studying autistic individuals and individuals in the BAP and the need for more research to be conducted on people who display autistic traits.

### 3.4. Impact of Co-Morbid Disorders on Well-Being

Highlighting the importance of considering co-morbidities when studying sleep quality in people with ASD or in the BAP, Stewart and colleagues (2020), in their cross-sectional study, asked 13,897 older adults between the ages of 50 and 81 to report their sleep problems and clinical depression and anxiety levels. One hundred and eighty-seven individuals showed high autistic traits in childhood and at the time of the study, and 6740 individuals showed no autistic traits [33]. These individuals were divided into two groups, and their scores were compared. Overall, they found a significant positive correlation between sleep problems, depression and anxiety and this correlation was similar between genders. They also found a positive correlation between age and nighttime and daytime sleep durations. The findings of this study are quite important as they highlight that individuals who display autistic traits often report having sleep problems.

Additionally, this study highlights that co-morbidities such as depression and anxiety contribute to poor sleep quality, suggesting a bi-directional relationship between the two. To add to this, Adams, Clark and Keen (2019) studied the relationship between anxiety and quality of life in 71 children between the ages of six and thirteen on the autism spectrum. They used the Social Communication Questionnaire (SCQ) that helps the participants’ parents to indicate the presence of social, communicative and stereotyped behaviours, along with the Anxiety Scale for Children with Autism Spectrum Disorder- Child (ASC-ASD-C), which is divided into four subscales, namely performance anxiety, anxious arousal, separation anxiety and uncertainty [34]. 57.7% of children reported having elevated anxiety along with a lower Health-related quality of life (HRQoL). Anxious arousal was a significant predictor of physical functioning, i.e., participants with higher anxious arousal symptoms had poorer physical functioning HRQoL scores. Overall, they concluded that individuals on the autism spectrum are more likely to experience anxiety symptoms and that they may experience anxiety differently from neurotypical individuals. This suggested that people with ASD, or autistic traits, may experience anxiety symptoms that impact their quality of life.

When considering other co-morbidities, Kwon, Kim and Kwak (2019) studied the relationship between sleep quality, ADHD symptoms and quality of life in 195 college students in South Korea [35]. They found that total quality of life, which consisted of the same domains as the ones studied in [16], correlated negatively with ADHD symptoms, and poorer sleep quality was associated with more ADHD symptoms. Other factors that improved overall quality of life were high academic performance, economic status, and gender (male students reported having a better quality of life). When looking at the specific domains, ADHD symptoms accounted for some variance in Physical, Psychological and Environmental Quality of Life but not Social Quality of Life. On the other hand, sleep quality accounted for variance in physical, psychological, and social quality of life but not environmental quality of life. Overall, a lack of sleep decreased the students’ ability to remain attentive and learn and control emotions, triggering impulsive behaviours.

Further highlighting the importance of considering ADHD when studying ASD, Smith, Garcha and James (2023) studied the association between autistic and ADHD traits in 155 secondary school students in Wales and how these traits impact well-being. They found a significant correlation between autistic and ADHD traits using the Autism Spectrum Quotient (ASQ) and ADHD scale, respectively, in traits such as hyperactivity and peer problems [36]. However, only autistic traits were significantly associated with conduct problems and conduct problems were negatively associated with positive coping. Only ADHD traits were significantly negatively associated with prosocial behaviour, and prosocial behaviour was significantly associated with gender (specifically females) and negative coping. Similarly, Antshel and Russo (2019) found in their umbrella review of studies conducted on youths that the hyperactive-impulsive symptom in ADHD was strongly correlated with restricted and repetitive behaviours in ASD and vice versa, often causing difficulties in social interactions and executive functioning in both [37]. This highlights the importance of considering ADHD traits when studying autistic traits and how this impacts well-being.

Furthermore, the scale used in the above studies [27,33,36,37] to detect autistic traits, the ASQ, only consisted of ten items (shortened to five items in Stewart and colleagues’ study), assessed their socio-communicative traits in childhood and at the time of the study and had a binary answering system (yes or no). Although the researchers reported that the scale had good specificity and sensitivity, the scale does not provide a good representation of the spectrum of traits that autistic people and those that fall under BAP may present as socio-communicative traits are one of many [2]. Hence, it is desirable to use the Comprehensive Autistic Traits Inventory (CATI) to improve on the general nature of most other measures of autistic traits, such as the ASQ and BAPQ. CATI explores all the main dimensions that are associated with autism, such as cognitive rigidity and social camouflage, while reflecting additions made to the diagnostic criteria for ASD, such as sensory sensitivity, in the Diagnostic and Statistical Manual of Mental Disorders fifth edition (DSM-5) [2,38]. This highlights the scale’s applicability in identifying autistic traits and diagnosing ASD, as opposed to ASQ and BAPQ, in accordance with the revised version, DSM-5-TR [4].

### 3.5. Other Factors Impacting Sleep Quality and Autistic Traits

One factor that has shown up a considerable number of times is the impact of smartphone and internet use on sleep quality and its correlation with ASD or autistic traits. Four studies (three in China and one in the U.K.) investigated the impact of smartphone use and addiction on the mental health and well-being of autistic students; three studies explored its impact on sleep quality. One explored its correlation with depressive symptoms and one with anxiety and loneliness. In general, all studies reported poorer sleep quality, more depressive symptoms and higher levels of loneliness and social interaction anxiety [39,40,41,42]. Along with this, the study by Huang and colleagues (2020) reported poorer physical health, and males reported having worse sleep quality than females [40]. Normand and colleagues (2021) also conducted a systematic review to explore the impact of problematic internet use (PIU) over different age groups and reported a correlation between PIU and depression, inattention, hyperactivity, impulsivity, opposition and escapism [43].

## 4. Conclusions

Overall, the small literature justifies further research on autistic traits, sleep quality, and the well-being of university students. While the review highlights the associations between autistic traits and well-being, autistic traits and sleep quality and sleep quality, and the differences in correlation among different age groups, gender and cultural backgrounds, no study explores the correlation between all three variables in university students. Neurotypical and neurodivergent university students have been found to have a decline in their well-being upon starting university, with neurodivergent students struggling more due to sensory problems and feelings of isolation, which is a result of their disability. Studies explore the factors that impact sleep quality and well-being, as well as the impact of autistic traits on well-being in university students. Only a handful of studies explore a link between autistic traits and sleep quality, specifically in terms of fatigue and sleep problems.

Some of the strengths of this review are that it is flexible yet provides sufficient depth in the interpretation. However, due to its flexibility, replicability is difficult. Each narrative review is dependent on the author’s choice of screening, sampling and analysis method [15]. While the review shows a selective inclusion criteria, it is difficult to evaluate the quality of the review with a strict criterion. The review identifies the need for research in this field and features that need to be adapted or changed in future research. Our first empirical study will have the following features: large sample size, appropriate measures such as CATI, ADHDQ, SDQ WPQ, sleep questionnaire and an appropriate multivariate and modelling analysis strategy. Such a study will form the basis for further potential studies using additional measures, a longitudinal design, and qualitative analyses to aid in the interpretation of the findings.

## Figures and Tables

**Table 1 healthcare-12-02027-t001:** Well-being process model.

	Positive	Negative
General Characteristics	Social support	Demands
Appraisals	Life satisfaction	Perceived stress
Individual characteristics	Positive personality	Negative coping
Outcomes	Happiness	Anxiety and Depression

## Data Availability

No new data were created or analysed in this study. Data sharing is not applicable to this article.

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
