# Peer review of "Exploring the Associations between Autistic Traits, Sleep Quality and Well-Being in University Students: A Narrative Review"

_healthcare, 2024, doi:10.3390/healthcare12202027_

Round 1
Reviewer 1 Report
Comments and Suggestions for Authors
Introduction
Line 32: Further discuss the diagnosis (for example, I suggest you to use the following reference: https://doi.org/10.61838/kman.aftj.5.3.8)
Line 65: You should provide appropriate reference for statistical reports.
2. Methods and Materials
I suggest to further explain this section by dividing it into some subsections like study design, search strategy, data analysis method, etc.
3. Results
I highly recommend to use subsections for your results to make it more readable.
4. Conclusion
This part is too short. You must discuss the findings and make suggestions for future research and implications.

Author Response
We thank the reviewers for their constructive comments. Our response to these is shown below and the changes made to the revised manuscript,
Introduction
Line 32: Further discuss the diagnosis (for example, I suggest you to use the following reference: https://doi.org/10.61838/kman.aftj.5.3.8)
Further discussion is now included.
Line 65: You should provide appropriate reference for statistical reports.
A reference has now been provided.
- Methods and Materials
I suggest to further explain this section by dividing it into some subsections like study design, search strategy, data analysis method, etc.
The methods section has been divided into subsections and more information on analysis strategy has been provided.
- Results
I highly recommend to use subsections for your results to make it more readable.
The results section has been divided into subsections.
- Conclusion
This part is too short. You must discuss the findings and make suggestions for future research and implications.
The conclusion now summarises the information discussed in the paper and explore some of the strengths and limitations, along with explaining how the findings of the paper will be used in our first empirical study.
Reviewer 2 Report
Comments and Suggestions for Authors
Autistic Traits, Sleep Quality and Well-being in University 2 Students: A Narrative Review
I appreciate your interesting study. I have some suggestions that could enhance the comprehensiveness of this research.
Materials and Methods
l Which languages of research are included in the search? Kindly provide additional information.
l Have you searched information from other trustworthy databases like MEDLINE, Embase, and PsycINFO?
l Kindly provide additional information regarding the search selection that was obtained. In what ways is it excluded? Ultimately, what was the remaining amount of research that could be applied to this study? Suggest additional presentations in the figure.
l Please provide further details on the process of data extraction? Additionally, it would be helpful if you could include a table to enhance the clarity of the explanation.
Others To enhance the comprehensiveness of this research. Kindly include the following points:
- Strengths and limitations
- Comparison with existing literature
- Implications for practice and research
Author Response
We thank you for your constructive comments. Our response to these is shown below and the changes made to the revised manuscript,
Materials and Methods
l Which languages of research are included in the search? Kindly provide additional information.
This has been added under the ‘design’ subsection of the methods.
l Have you searched information from other trustworthy databases like MEDLINE, Embase, and PsycINFO?
Yes, a preliminary search was done on PubMed. This has been added under the ‘design’ subsection of the methods.
l Kindly provide additional information regarding the search selection that was obtained. In what ways is it excluded? Ultimately, what was the remaining amount of research that could be applied to this study? Suggest additional presentations in the figure.
This is a narrative review and what you are suggesting would be appropriate for a systematic review but not a narrative one.
l Please provide further details on the process of data extraction? Additionally, it would be helpful if you could include a table to enhance the clarity of the explanation.
As above.
Others To enhance the comprehensiveness of this research. Kindly include the following points:
- Strengths and limitations
These are now addressed in the literature.
- Comparison with existing literature
There is no specific literature addressing this topic within the context of the well-being model used here.
- Implications for practice and research
The conclusion now summarises the information discussed in the paper and explore some of the strengths and limitations, along with explaining how the findings of the paper will be used in our first empirical study.
Reviewer 3 Report
Comments and Suggestions for Authors
Geeral comment -The title of the article "Autistic Traits, Sleep Quality and Well-being in University Students: A Narrative Review" seems to give the point out the relationships between the variables: autistic traits, sleep quality and well-being in univeristy students based on the narrative review. However the article is based on the narrative description of the reviwed literature there is nothing given about the relationships but only about the separated variable. The title suggests one thing and the content of the article is about something else. This is very disappointing for the reader. A publication presented in a high-scoring journal should meet the criteria of novelty, a creative approach to the issue, and precise answers to the questions asked.
Introduction - there are given the general information about ASD and comordities and well-being. The theoretical backgound based on the Demand-Resources-Individual-Effects (DRIVE) model is not well justified. Usually this model is used as the ground for work stress and burn-out. Please specify the adequate arguments and convincing justifications or references to data from the literature. It is not specified whether the table 1 content refers to the original constructs resulting from the adopted model or is the result of the authors' creativity.
Method - please be mo specific in describing the data collecting procedure and criteria. It is narrative review so this part of the article might be not so respective as in systematical revie however it would be professional to be more precise.
Results -First of all, it is worth mentioning all the articles that meet the assumed inclusion criteria and then discussing them in details. Please be more specific in presenting the methods of reviewed articles. If I understand well - a presentation scheme was adopted by authors like that: the size of the donor sample and assessmnet instruments used was given based on the original articles. This may not be sufficient because the specific characteristics of the subjects or the procedure used may have a specific impact on the results. Narrative review, especially if there are not many works cited/analyzed ( as it is in the article), is a good context for an in-depth analysis of possible variables moderating/mediating the observed dependencies. It would be good if the authors would be willing to explain to the reader the procedural intricacies of the cited research projects, not just the basic results. The article version submitted for review forces the reader to look for answers by directly reading the source materials, which reduces the scientific value of the article.
Discussion - results, the description of the results of reviewed articles, are presneted however there is the lack of discussion.
Conclussions -are very laconic and consist only in listing variables that can be included in subsequent research. They do not refer to critical analysis and more in-depth arguments allowing for drawing conclusions regarding not only the size of the studied samples or variables, but also procedural or theoretical modifications. Please be more specific and critical.
Author Response
We thank you for your constructive comments. Our response to these is shown below and the changes made to the revised manuscript,
Geeral comment -The title of the article "Autistic Traits, Sleep Quality and Well-being in University Students: A Narrative Review" seems to give the point out the relationships between the variables: autistic traits, sleep quality and well-being in univeristy students based on the narrative review. However the article is based on the narrative description of the reviwed literature there is nothing given about the relationships but only about the separated variable. The title suggests one thing and the content of the article is about something else. This is very disappointing for the reader. A publication presented in a high-scoring journal should meet the criteria of novelty, a creative approach to the issue, and precise answers to the questions asked.
The title has been updated to reflect the contents of the paper.
Introduction - there are given the general information about ASD and comordities and well-being. The theoretical backgound based on the Demand-Resources-Individual-Effects (DRIVE) model is not well justified. Usually this model is used as the ground for work stress and burn-out. Please specify the adequate arguments and convincing justifications or references to data from the literature. It is not specified whether the table 1 content refers to the original constructs resulting from the adopted model or is the result of the authors' creativity.
A rationale for using DRIVE model in this paper has been discussed.
Method - please be mo specific in describing the data collecting procedure and criteria. It is narrative review so this part of the article might be not so respective as in systematical revie however it would be professional to be more precise.
The methods section has been divided into subsections and more information on analysis strategy has been provided.
Results -First of all, it is worth mentioning all the articles that meet the assumed inclusion criteria and then discussing them in details. Please be more specific in presenting the methods of reviewed articles. If I understand well - a presentation scheme was adopted by authors like that: the size of the donor sample and assessmnet instruments used was given based on the original articles. This may not be sufficient because the specific characteristics of the subjects or the procedure used may have a specific impact on the results. Narrative review, especially if there are not many works cited/analyzed ( as it is in the article), is a good context for an in-depth analysis of possible variables moderating/mediating the observed dependencies. It would be good if the authors would be willing to explain to the reader the procedural intricacies of the cited research projects, not just the basic results. The article version submitted for review forces the reader to look for answers by directly reading the source materials, which reduces the scientific value of the article.
Discussion - results, the description of the results of reviewed articles, are presneted however there is the lack of discussion.
Since this is a narrative review, the studies are reviewed and discussed concurrently, as explained in Sukhera (2022).
Conclusions -are very laconic and consist only of listing variables that can be included in subsequent research. They do not refer to critical analysis and more in-depth arguments allowing for drawing conclusions regarding not only the size of the studied samples or variables, but also procedural or theoretical modifications. Please be more specific and critical.
The conclusion now summarises the information discussed in the paper and explore some of the strengths and limitations, along with explaining how the findings of the paper will be used in our first empirical study.
Round 2
Reviewer 3 Report
Comments and Suggestions for Authors
The article is succesfully improved in reference to previous comments